# Unraveling the Potential of Attentive Bi-LSTM for Accurate Obesity Prognosis: Advancing Public Health towards Sustainable Cities

**DOI:** 10.3390/bioengineering11060533

**Published:** 2024-05-23

**Authors:** Hina Ayub, Murad-Ali Khan, Syed Shehryar Ali Naqvi, Muhammad Faseeh, Jungsuk Kim, Asif Mehmood, Young-Jin Kim

**Affiliations:** 1Interdisciplinary Graduate Program in Advance Convergence Technology and Science, Jeju National University, Jeju 63243, Republic of Korea; hina1112@stu.jejunu.ac.kr; 2Department of Computer Engineering, Jeju National University, Jeju 63243, Republic of Korea; muradali@stu.jejunu.ac.kr; 3Department of Electronics Engineering, Jeju National University, Jeju 63243, Republic of Korea; syedshehryar@stu.jejunu.ac.kr (S.S.A.N.);; 4Department of Biomedical Engineering, College of IT Convergence, Gachon University, 1342 Seongnamdaero, Sujeong-gu, Seongnam-si 13120, Republic of Korea; jungsuk@gachon.ac.kr; 5Medical Device Development Center, Osong Medical Innovation Foundation, Cheongju 28160, Republic of Korea

**Keywords:** obesity prediction, deep learning, attention, multi-class classification, smart health of residents, smart living towards sustainable city development

## Abstract

The global prevalence of obesity presents a pressing challenge to public health and healthcare systems, necessitating accurate prediction and understanding for effective prevention and management strategies. This article addresses the need for improved obesity prediction models by conducting a comprehensive analysis of existing machine learning (ML) and deep learning (DL) approaches. This study introduces a novel hybrid model, Attention-based Bi-LSTM (ABi-LSTM), which integrates attention mechanisms with bidirectional Long Short-Term Memory (Bi-LSTM) networks to enhance interpretability and performance in obesity prediction. Our study fills a crucial gap by bridging healthcare and urban planning domains, offering insights into data-driven approaches to promote healthier living within urban environments. The proposed ABi-LSTM model demonstrates exceptional performance, achieving a remarkable accuracy of 96.5% in predicting obesity levels. Comparative analysis showcases its superiority over conventional approaches, with superior precision, recall, and overall classification balance. This study highlights significant advancements in predictive accuracy and positions the ABi-LSTM model as a pioneering solution for accurate obesity prognosis. The implications extend beyond healthcare, offering a precise tool to address the global obesity epidemic and foster sustainable development in smart cities.

## 1. Introduction

Over the last three decades, the mean body mass index (BMI; weight in kilograms divided by square height in meters) has increased worldwide by 0.4 kg m^−2^ per decade, which causes obesity [1]. Obesity is an abnormal condition in which excess fat accumulates in adipose tissue to the point of affecting health. Too many fat cells or increased body fat lead to obesity. As a result of the abundance of food consumed, sedentary lifestyles, and lack of physical activity, obesity is a complex issue [2,3]. Approximately 13% of the adult population worldwide was obese in 2016, according to the World Health Organization (WHO) [4]. A significant portion of the obese population is from younger generations since over 34 million children under 5 are overweight [5]. These alarming statistics illustrate that effective strategies are needed to stem obesity’s rising tide [6].

Many researchers accounted for a cohort effect for combined time and age effects by analyzing cross-sectional data on obesity prevalence [7]. Many researchers have adopted a holistic approach to obesity risk assessment, employing lifestyle, genetic, and dietary data [8,9,10]. Physiological and demographic factors were used to predict obesity risk in ref. [11]. Despite its impressive accuracy in initial trials, the model struggled to handle non-linear relationships within the dataset. As a result, obesity’s multifactorial etiology poses a challenge because complex interactions are required to capture them. Another study used neural networks, they developed intricate neural architectures for forecasting obesity onset, achieving commendable predictions [12]. Nevertheless, it could be limited in its application when limited data are available due to its reliance on substantial labeled training data. The field’s progress depends on balancing predictive power with data quality and practicality in practice.

There have been various approaches to obesity prediction, resulting in insights accompanied by inherent limitations [13]. Longitudinal studies incorporate lifestyle and genetic data for dynamic risk forecasting, but retrospective self-reporting introduces bias [14]. Ensemble models in machine learning enhance prediction performance, but it also demands robust data quality [15,16,17]. Real-time health monitoring is essential for risk assessment, yet continuous data streams and privacy concerns present challenges [18,19,20]. By encompassing holistic influences, mitigating biases, and striking a balance between innovation and practicality, these models reveal the complexity of obesity prediction. With the advent of neural networks, obesity prediction has gained a new dimension [21]. In addition to using intricate neural architectures for predicting obesity onset, Landscape pushed the boundaries of model complexity [22]. These models demonstrated a commendable level of predictive accuracy. Despite this, such architectures require a substantial amount of labeled training data, potentially limiting their utility in scenarios with limited access to data. Although these pioneering studies contribute to our understanding of obesity prediction, they also highlight certain limitations. There are inherent challenges associated with lifestyle changes, genetic influences, and their interaction within the obesity ecosystem due to their dynamic nature. Additionally, these models are susceptible to self-reported data biases. It becomes increasingly apparent that obesity prediction models need to be refined as the field advances. By considering both individual characteristics as well as societal factors, a comprehensive and integrated approach could yield more robust solutions.

In this paper, we propose a novel solution to address the pressing global issue of obesity by introducing the Attention-based Bi-LSTM (ABi-LSTM) model. The proposed ABi-LSTM model leverages the power of machine learning and deep learning to enhance obesity level prediction and significantly improve performance in terms of accuracy, precision, recall, and f1 score. With an accuracy of 96.5%, the ABi-LSTM model outperforms all existing state-of-the-art models, constituting a revolutionary breakthrough towards accurate obesity forecasting. Our research presents an innovative method for accurately predicting obesity levels, showcasing substantial improvements in predictive performance. The discoveries have far-reaching implications for public health and related research areas, offering a robust and precise mechanism to tackle the worldwide obesity crisis. Moreover, our investigation underscores the importance of comprehensive data collection through questionnaires and sensor data, enabling us to explore the intricate interactions among environmental influences, genetic effects, and lifestyle variables. Our study bridges the gap between healthcare and urban planning, illuminating the potential of data-driven techniques to promote healthy living in urban environments in a world that is fast evolving and where the idea of smart cities is gaining traction. Additionally, the hybrid technique we presented makes a number of noteworthy additions, which are enumerated as follows:Introduction of the Attention-based Bi-LSTM (ABi-LSTM) model, achieving a remarkable accuracy of 96.5% in obesity prediction.Advancements in predictive accuracy surpass existing models, offering a superior tool for obesity prognosis.Significance for public health and healthcare systems, addressing the global obesity epidemic with a precise and robust solution.Emphasis on comprehensive data collection, utilizing surveys and sensor data to capture the complex interactions between lifestyle, genetics, and environmental factors.Bridging the gap between healthcare and urban planning in the context of smart cities, offering insights into promoting healthier living within urban environments.

## 2. Related Work

Obesity prediction has evolved with various methodologies, addressing its intricate nature. In the early stages of research, statistical approaches were employed, as demonstrated by Anderson et al. [23] in their meta-analysis of long-term weight loss trends. A further breakthrough was made by Jiménez-Santos et al. in federated learning for secure medical data sharing [24]. The inclusion of genetic and non-genetic attributes has enriched predictive models. Using electronic health records, Chu et al. [25] identified key risk factors using decision trees. The researchers combined genetic markers with dietary habits to improve prediction accuracy by using random forests. Network-based strategies were developed using multi-omics data to construct obesity-related interaction networks [26].

We investigated temporal trends in predictors [27] using recurrent neural networks (RNNs) for time series analysis. A natural language processing approach was used to identify obesity-related language patterns in unstructured clinical text by Seddik et al. [28]. Choi et al. [29] incorporated an attention mechanism into deep learning for interpretability. Hybrid models, such as Bhavya et al.’s [30] ensemble model that combines support vector machines with random forests to predict disease, received attention. Using domain knowledge, Pan et al. [31] enhanced predictive capabilities by integrating deep learning with domain knowledge. In addition, in [32,33,34], the authors introduced deep learning models based on RNNs to enhance the prediction performance by investigating temporal data patterns for sustainable city development. Jain et al. [35] also used a random forest approach to optimize prediction accuracy by selecting genetic features. Wang et al. [36] presented the FL-STNet model, leveraging the Swin-Transformer network with focal loss for identifying pathological subtypes of lung adenocarcinoma. It demonstrates superior accuracy in classifying lung adenocarcinoma subtypes compared to pathologists, with an average classification accuracy of 85.71%. In [37,38], the authors investigated different parameters towards sustainable city development. Similarly, in [39], the authors attempted to determine a monotonic relationship between temporal parameters. Different analyses are employed in [40] to investigate the seasonal coherence in temporal data for sustainable city development. In [41], Huang et al. proposed the LA-ViT model for grading laryngeal squamous cell carcinoma (LSCC) based on histopathological images. It employs transformers constrained by learned-parameter-free attention to enhance interpretability and reduce the proportion of low-effect background semantic information, improving accuracy in LSCC grading.

Interdisciplinary research has flourished as a means of predicting obesity more comprehensively. From genetic and metabolic data, Watanabe et al. [42] developed a deep learning model to predict body mass index. Feretzakis et al. [43] explored domain-specific data to enhance classification performance. Using deep learning, they predicted nosocomial infections. Multimodal approaches have also gained traction. The multimodal deep learning model developed by Yoo et al. [44] integrates genetic, clinical, and behavioral data. Marcos et al. [45] examined how genetic and dietary factors interact in interaction networks. By leveraging textual information, they could predict obesity using natural language processing. Using ensemble machine learning, Zhang et al. [46] enhanced obesity risk assessment by including genetic and lifestyle factors. Lin et al. [47] investigated genotypic and phenotype information integration with multi-omics data. Khan et al. [48] applied automated ML on multi-source data in response to the advent of big data of patients rehabilitation. In addition, Ven et al. [49] used multidimensional models to predict obesity by combining genetic, clinical, and environmental factors. Similarly, in [50,51], the authors introduced a hybrid model by combining U-Net and spatial transformation network to enhance prediction using temporal data. In [52], the authors used a Monte Carlo-based analysis method to analyze spatially distributed data for enhancing sustainable city development. Using multi-source data, Safaei et al. [53] enhanced the accuracy of obesity risk prediction by integrating deep learning.

Machine learning techniques were used by Siddiqui et al. [54] to predict early obesity based on longitudinal data. Callens et al. [55] combined random forests with gradient boosting to improve prediction accuracy. Through the integration of wearable device data, Gholamhosseini et al. [56] developed a model for assessing obesity risk in real time. Social network analysis shows Nam et al. [57] could predict obesity based on social interactions and behavior patterns. Si et al. [58] proposed a transfer learning approach to account for varying data distributions.

According to Jiang et al. [59], the interdisciplinary exploration extended to image-based prediction based on medical images. In addition, Chong et al. [60] used graph-based models to capture the complex relationships between genetic markers and obesity. A deep learning model for obesity prediction based on explainable AI was introduced by Gupta et al. [61]. Huang et al. [62] also focused on machine learning interpretability when selecting features and explaining models in obesity prediction. As a result, obesity prediction research has embraced diverse methodologies, each bringing a unique perspective to its multifaceted nature. This field is characterized by the synergy of several techniques, including traditional statistics, deep learning, network analysis, image analysis, and interpretability methods.

## 3. Methodology

Obesity is a growing issue on a global and local level that transcends socioeconomic and geographic barriers. Obesity is seen as a serious public health issue and is linked to many annual deaths. It is important to look into factors contributing to obesity, such as insufficient exercise, seasonal work-hour differences, and diminishing activity. Furthermore, advanced techniques can be employed to analyze the cohort effect of obesity over time for enhanced evaluation. In this research, advanced deep learning techniques, such as bidirectional long short-term memory (Bi-LSTM) models with attention mechanisms, are used to forecast and analyze obesity levels. By incorporating attention mechanisms, our approach allows for a deep exploration of temporal patterns and cohort effects associated with obesity, offering valuable insights for public health interventions and policy formulation. In this section, the methodology unfolds in three parts: starting with the data description and preprocessing, followed by an examination of the causes and effects of obesity. The section concludes by detailing the proposed framework, the Attention-based Bi-LSTM (Abi-LSTM), along with the functionality of the attention layer.

### 3.1. Data Description and Preprocessing

The obesity dataset utilized in our study is structured and tabular in nature. It comprises multiple attributes representing various aspects of individuals’ eating behaviors, physical conditions, and demographic information. Each row in the dataset corresponds to an individual, while each column represents a specific feature or attribute.

The dataset contains a mix of categorical, numerical, and textual data. Categorical data include variables such as gender, mode of transportation, and consumption of high-calorie food, which are represented as discrete categories. Numerical data include features such as age, weight, and frequency of physical activity, represented as numerical values. Textual data include qualitative information or descriptions of certain features, such as dietary habits or lifestyle choices.

Before model training, we conducted extensive data preprocessing steps to clean and prepare the dataset for analysis. This included handling missing values through imputation techniques, encoding categorical variables using methods like one-hot encoding, scaling numerical features to a standard range, and performing feature engineering to extract relevant information from the raw data.

Table 1 provides a comprehensive summary of the dataset attributes categorized by eating habits, physical condition, and other variables, along with detailed descriptions of their meanings and significance in obesity prognosis. Each feature in the dataset holds significance in understanding and predicting obesity. Attributes related to eating habits, physical conditions, and demographic information provide valuable insights into individuals’ lifestyles and health statuses. For instance, frequent consumption of high-calorie foods, physical activity frequency, and transportation choices are indicative factors influencing obesity risk.

These features work together to forecast the prevalence of obesity (NObeyesdad) amongst the individuals in the dataset. Using these data, we harness these characteristics to delve into and create an Abi-LSTM predictive model that unveils the complex interplay between lifestyle elements, physical well-being, and the probability of obesity. This dataset serves as a valuable tool for gaining insights into the various factors that influence obesity, essential for crafting effective interventions and strategies in public health.

### 3.2. Causes and Effects of Obesity

A strong link exists between obesity and health, well-being, and society’s equilibrium, underscoring the need for comprehensive understanding and intervention [57,63]. Despite obesity’s burgeoning prevalence, little research has been conducted to examine its impact on employee engagement, safety, and productivity. This knowledge gap is particularly pronounced in the context of physically demanding and time-sensitive big data labor. Due to physical limitations, obesity could possibly affect the ability to schedule, impede, or postpone work activities, which warrants a thorough investigation. There are multiple dimensions to obesity’s impact, including food supply, economic stability, and community vitality, as shown in Figure 1. It is imperative to conduct comprehensive research to understand the repercussions of obesity on individuals’ ability to contribute effectively to the workforce. Physically demanding tasks and strict deadlines are inherent in extensive data work. Understanding how obesity impacts work efficiency, safety, and productivity requires rigorous exploration. By recognizing the effects of obesity, initiatives can be launched to promote healthier habits, facilitating weight loss or maintenance. These insights can also be used to develop comprehensive worker health programs, enhancing the industry’s capacity to address obesity-related issues. Considering obesity in the context of work procedures can provide opportunities to redesign tasks to make them safer and more efficient for obese workers. Taking this approach contributes to fostering a culture of inclusion at work. As a result, an in-depth analysis of obesity’s multifaceted effects is imperative. Research on obesity, workforce productivity, and safety can contribute to societal well-being, economic stability, and the optimization of work processes.

### 3.3. An Overview of the Proposed Model

This section thoroughly describes the proposed model, moving from the initial raw data through several crucial preprocessing processes, as shown in Figure 2. The first point of the voyage is the raw data, which form the basis of our investigation. We use a multi-step preparation method to enhance the quality and usability of these data. In this pipeline, irrelevant attributes are removed, categorical-to-numeric transformation is used to handle non-numeric data, missing values are imputed to complete the data, the best features are chosen for high performance and low computational cost, and normalization is used to ensure uniform scaling throughout the dataset. Together, these preprocessing procedures set up the data for analysis and model training, constituting a critical first stage in our study.

Once the data are preprocessed, they are channeled into our proposed Bi-LSTM with attention model, a cornerstone of our research. Simultaneously, we compare the same prepared data to other state-of-the-art deep neural network (DNN) models, including CNN, RNN, LSTM, Bi-LSTM, and TabNet. This comprehensive set of models forms the basis of our comparative analysis. In the following “Comparison Analysis” step, we carefully evaluate and compare the results produced by each model. This thorough comparison demonstrates the benefits of our proposed ABi-LSTM with a focus on design and offers insightful information about how it performs in contrast to leading-edge models. These findings greatly influence our conclusions, which demonstrate how well our suggested approach works when applied to actual problems.

### 3.4. Proposed Framework

In this research, we introduce a novel framework designed for multilabel classification tasks, leveraging the capabilities of a Bidirectional Long Short-Term Memory (Bi-LSTM) network enhanced with an attention mechanism. We utilized ABi-LSTM, which excels in processing data sequences and is suitable for tasks involving sequential data, such as our obesity level prediction problem. As a variant of a recurrent neural network (RNN), the ABi-LSTM model is well-suited to handle sequential data, like the obesity dataset we used. The ability to proficiently capture temporal dependencies and patterns within the data is crucial for comprehending the intricate linkages that exist between obesity levels and lifestyle factors throughout time. This framework suits scenarios where each input instance can be associated with multiple labels. The architecture of our model begins with data preprocessing, where the input data are appropriately reshaped for compatibility with the subsequent layers. We then employ three successive Bidirectional LSTM layers, each serving a unique purpose. The first Bi-LSTM layer, utilizing the rectified linear unit (ReLU) activation function, captures initial patterns in the data. The second Bidirectional LSTM layer, employing the hyperbolic tangent (tanh) activation function, further refines these patterns, followed by a third Bidirectional LSTM layer with a similar activation function to capture nuanced dependencies.

In the proposed study, the attention layer dynamically assigns weights to different input features based on their relevance to predicting obesity. The attention mechanism focuses more on certain features such as height, weight, and physical activity levels while making predictions about obesity. The attention mechanism allows the model to weigh these features differently for each input sample, enhancing the model’s ability to capture complex relationships within the data. A soft attention mechanism calculates the attention weights using a learned function that considers the similarity between the current input and the context vector. The attention weights are then applied to the output of the Bi-LSTM layer to produce a context vector, which is used for making predictions. Soft attention mechanisms are effective for tasks where different parts of the input sequence contribute unequally to the output. The details of the parameters used in the experiments are listed in Section 4.4.

The attention-based Bi-LSTM model’s parameter selection is essential for multilabel classification, especially when predicting obesity. Every parameter has a unique effect on the design and behavior of the model, which directly affects how well it can handle the complexity of our data. By specifying the input dimension, you can make sure that the model can handle the characteristics—like height, weight, and degree of physical activity—that are important for predicting obesity. Because of its bidirectional architecture, the model can record relationships in both forward and backward directions, which makes it easier to comprehend the input sequence in its entirety. The hidden dimension and number of layers are two other factors that affect the model’s ability to identify complex patterns and correlations in the data. By allowing it to concentrate on the most informative components of the input sequence, the attention mechanism improves prediction accuracy and the model’s performance. In general, elaborating on these factors offers a significant understanding of how our model is customized to handle the intricacies of multilabel classification jobs, ultimately leading to more precise and dependable obesity predictions.

An essential addition to our framework is the Attention layer, which dynamically weights the outputs of the LSTM layers, focusing on the most informative elements within the input sequence. This attention mechanism enhances the model’s ability to make precise predictions. Finally, the output layer employs a Dense layer with sigmoid activation, producing a probability vector for each label, where each element signifies the likelihood of the respective label’s presence in the input. A summary of the proposed framework is illustrated in Figure 3.
$$\begin{array}{cc} \mathrm{Forward}\;\mathrm{LSTM}:\\ i_{t} & =\sigma (W_{xi}\cdot x_{t}+W_{hi}\cdot h_{t-1}+W_{ci}\cdot c_{t-1}+b_{i})\\ f_{t} & =\sigma (W_{xf}\cdot x_{t}+W_{hf}\cdot h_{t-1}+W_{cf}\cdot c_{t-1}+b_{f})\\ g_{t} & =\tan h(W_{xg}\cdot x_{t}+W_{hg}\cdot h_{t-1}+b_{g})\\ o_{t} & =\sigma (W_{xo}\cdot x_{t}+W_{ho}\cdot h_{t-1}+W_{co}\cdot c_{t-1}+b_{o})\\ c_{t} & =f_{t}⊙c_{t-1}+i_{t}⊙g_{t}\\ h_{t} & =o_{t}⊙\tan h\left(c_{t}\right) \end{array}$$
$$\begin{array}{cc} \mathrm{Backward}\;\mathrm{LSTM}:\\ i_{t}^{\prime } & =\sigma (W_{xi}^{\prime }\cdot x_{t}+W_{hi}^{\prime }\cdot h_{t+1}^{\prime }+W_{ci}^{\prime }\cdot c_{t+1}^{\prime }+b_{i}^{\prime })\\ f_{t}^{\prime } & =\sigma (W_{xf}^{\prime }\cdot x_{t}+W_{hf}^{\prime }\cdot h_{t+1}^{\prime }+W_{cf}^{\prime }\cdot c_{t+1}^{\prime }+b_{f}^{\prime })\\ g_{t}^{\prime } & =\tan h(W_{xg}^{\prime }\cdot x_{t}+W_{hg}^{\prime }\cdot h_{t+1}^{\prime }+b_{g}^{\prime })\\ o_{t}^{\prime } & =\sigma (W_{xo}^{\prime }\cdot x_{t}+W_{ho}^{\prime }\cdot h_{t+1}^{\prime }+W_{co}^{\prime }\cdot c_{t+1}^{\prime }+b_{o}^{\prime })\\ c_{t}^{\prime } & =f_{t}^{\prime }⊙c_{t+1}^{\prime }+i_{t}^{\prime }⊙g_{t}^{\prime }\\ h_{t}^{\prime } & =o_{t}^{\prime }⊙\tan h\left(c_{t}^{\prime }\right) \end{array}$$

Attention mechanism:$$\begin{array}{cc} e_{t} & =\tan h(W_{a}\cdot h_{t}^{*}+b_{a})\\ \alpha_{t} & =\frac{\exp(e_{t})}{\sum_{k=1}^{T}\exp\left(e_{k}\right)}\\ c_{t} & =\sum_{k=1}^{T}\alpha_{k}h_{k}^{*}\\ y_{t} & =\tan h(W_{c}\cdot c_{t}+W_{s}\cdot s_{t}+b) \end{array}$$

Here, σ denotes the sigmoid activation function, tanh denotes the hyperbolic tangent activation function, W and b are weight matrices and bias vectors, ⊙ represents element-wise multiplication, $h_{t}$ and $h_{t}^{\prime }$ are the cell and hidden states of the forward and backward LSTMs, $h_{t}^{*}$ is the concatenated hidden state, $e_{t}$ represents the energy score, $\alpha_{t}$ is the attention weight, $c_{t}$ is the context vector, and $y_{t}$ is the attention output.

To evaluate the effectiveness of the proposed framework, we adopt the Adam optimizer for training. The Adam optimizer’s efficiency in optimizing deep neural networks complements our model’s architecture. Experimental evaluation involves fitting the model to the training data and assessing its performance using various metrics, including accuracy, precision, recall, and F1-score. Throughout this paper, we visually represent our framework to help readers grasp its architecture intuitively. Furthermore, we present the results of our experiments, highlighting the framework’s ability to achieve accurate multilabel predictions. Combining Bidirectional LSTMs and attention mechanisms, our proposed approach demonstrates promising potential in tackling complex multilabel classification tasks across different domains.

## 4. Experimental Results and Performance Analysis

This section provides an overview of the implementation environment, evaluation metrics, and the experimental results obtained from the proposed ABi-LSTM model, designed for obesity level classification.

### 4.1. Experiment Environment

This section presents an overview of our advanced preprocessing pipeline tailored for supervised regression tasks. We provide a summary of the essential tools and technologies utilized in processing obesity classification data, as outlined in Table 2. Our primary programming language for implementing these experiments is Python. We leverage critical Python libraries such as Sklearn, Keras, TensorFlow, and Seaborn to facilitate our data processing pipeline. The entire process is meticulously designed and executed using Python.

Furthermore, we employ the Microsoft Comma Separated Values (CSV) format to support our work on classification tasks to store the original obesity data and house the processed data. This format enhances compatibility and accessibility, ensuring that our data are readily available and well-suited for classification analysis.

### 4.2. Algorithm for the Proposed Model

In Algorithm 1, we divide obesity level classification into two primary phases. In the first phase, data preprocessing is conducted. This phase involves the removal of irrelevant attributes, the transformation of non-numeric data into a numeric format, the imputation of missing values, the selection of optimal features, and normalization to ensure uniform scaling. These steps collectively prepare the raw obesity dataset for further deep analysis.

In the second phase, multiple deep neural network (DNN) models, including CNN, RNN, LSTM, Bi-LSTM, TabNet, and the proposed model, are trained on the preprocessed dataset and evaluated using various performance metrics. The highest-performing model is then determined through an in-depth study, which also analyzes how each model stands up against the most effective in terms of efficiency. The outcomes of this comparison evaluation provide significant insight on the benefits of the suggested ABi-LSTM architecture, especially in resolving real-world issues with obesity level classification. The aforementioned algorithm ensures a logical and well-defined sequence of actions throughout the process by operating as a structured and systematic foundation for the research direction.
**Algorithm 1** Obesity Level Classification Pipeline
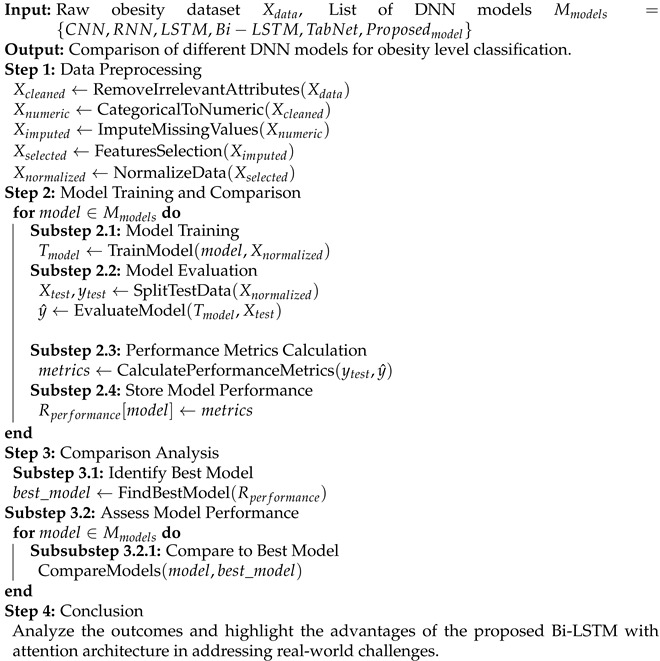


### 4.3. Evaluation Metrics

We rely on the usefulness of a confusion matrix to thoroughly assess the performance of our model. This matrix is a crucial tool for evaluating classification outcomes and also allows us to calculate important performance metrics like accuracy, precision, recall, and the F1 score. These metrics are crucial for measuring how effectively the proposed model categorizes instances into different classes.

The confusion matrix is a systematic representation that enables a deeper understanding of the model’s classification outcomes. Figure 4 exhibits the multi-class confusion metrics:

True Labels represent the actual class labels. Predicted Labels represent the class labels predicted by our model. At the top of the confusion matrix, we have seven classes in the target column.

Accuracy is a fundamental metric for assessing the correctness of our model’s predictions across all classes. It is calculated as the ratio of correctly predicted instances (TP and TN) to the total number of instances. The formula for Accuracy is as follows:(1)$$Accuracy=\frac{1}{N}\sum_{i=1}^{N}I\left(y_{i}={\hat{y}}_{i}\right)$$

Here is an explanation of the components of the formula:*N*: Total number of samples in the dataset.$y_{i}$: The true class label for the *i* -th sample.${\hat{y}}_{i}$: The predicted class label for the *i*-th sample.$I(y_{i}={\hat{y}}_{i})$: An indicator function that returns 1 if $y_{i}$ is equal to ${\hat{y}}_{i}$ (i.e., if the true label matches the predicted label) and 0 otherwise.

This formula calculates the accuracy by summing up the indicator function values for all samples and dividing by the total number of samples.

Equation (Equation 2) represents the standard formula for precision. Precision is a measure of the accuracy of the positive predictions made by a classification model. It quantifies the model’s ability to correctly identify relevant instances out of all instances predicted as positive. Precision can be interpreted as the probability that a positive prediction made by the model is indeed correct. Higher precision values indicate fewer false positives, which means the model is more reliable in identifying positive instances. Precision is particularly useful in cases where the cost of false positives is high, and it complements other performance metrics, such as recall and F1-score, in evaluating the overall effectiveness of a classification model.
(2)$$\mathrm{Precision}=\frac{TP}{TP+FP}$$
where:TP (True Positives) represents the number of instances correctly classified as positive;FP (False Positives) represents the number of instances incorrectly classified as positive.

Precision can be interpreted as the probability that a positive prediction made by the model is indeed correct. Higher precision values indicate fewer false positives, which means the model is more reliable in identifying positive instances. Precision is particularly useful in cases where the cost of false positives is high, and it complements other performance metrics such as recall and F1-score in evaluating the overall effectiveness of a classification model.

Recall measures the model’s ability to identify all positive instances correctly for a specific class out of all actual positive instances for that class. It quantifies the model’s ability to capture relevant instances and is particularly useful when the cost of false negatives is high.

The standard formula for recall is:(3)$$\mathrm{Recall}=\frac{TP}{TP+FN}$$
where:TP (True Positives) represents the number of instances correctly classified as positive;FN (False Negatives) represents the number of instances incorrectly classified as negative when they are actually positive.

Equation (Equation 3) represents the standard formula for recall. Recall measures the ability to identify relevant instances correctly. Higher recall values indicate fewer false negatives, meaning the model is better at capturing all positive instances. Recall is essential when missing positive instances can have severe consequences, such as in medical diagnosis or fraud detection. It complements other performance metrics such as precision and F1-score in evaluating the overall effectiveness of a classification model.

The F1 score is a metric that combines both precision and recall into a single value, providing a balanced assessment of a model’s performance. It quantifies the harmonic mean of precision and recall, giving equal weight to both metrics. The F1 score is particularly useful when there is an uneven class distribution or when false positives and false negatives have different consequences.

The standard formula for the F1 score is:(4)$$F1\;\mathrm{Score}=\frac{2\cdot \mathrm{Precision}\cdot \mathrm{Recall}}{\mathrm{Precision}+\mathrm{Recall}}$$
where:Precision is the precision of the model, as defined earlier;Recall is the recall of the model, as defined earlier.

Equation (Equation 4) represents the standard formula for the F1 score. It balances precision and recall, providing a single metric for model performance across all classes. A higher F1 score indicates better overall performance, with values closer to 1 indicating a better balance between precision and recall. The F1 score is commonly used in binary classification tasks but can also be extended to multi-class classification by taking the mean of F1 scores for each class.

These evaluation metrics, derived from the confusion matrix, provide a comprehensive understanding of our model’s performance, aiding in decision making and optimization across various domains and class labels.

### 4.4. Experimental Results and Analysis

In this comprehensive research study, we have undertaken a series of diverse experiments employing the cutting-edge ABi-LASTM model in conjunction with conventional DL models. These experiments are meticulously designed and conducted utilizing the Obesity Levels & Lifestyle dataset as our primary data source [64]. Furthermore, we have rigorously applied various data preprocessing techniques to optimize the dataset before feeding it into the selected models for a thorough and insightful comparison. Additionally, the dataset comprises 2.1 k instances, and we employ a train–test split, where 70% of the data are used for the training of the proposed model and 30% are used for testing.

Table 3 shows a configuration setup of our proposed model.

Additionally, the dataset comprises 10.5 k instances, and we employ 10-fold cross-validation for robust evaluation. The proposed approach is evaluated extensively, with detailed descriptions of the training and testing data provided. Specifically, the training data are processed with various data preprocessing techniques before being fed into the selected models, ensuring a thorough and insightful comparison. Moreover, the evaluation of the proposed approach includes metrics such as accuracy, precision, recall, and F1-score, providing a comprehensive understanding of model performance across different obesity levels and lifestyle factors.

#### 4.4.1. Analysis of Results Using Confusion Matrices

The confusion matrices shown in Figure 5 provide clear evidence of the proposed ABi-LSTM model’s superiority over other models in predicting obesity levels. The ABi-LSTM model consistently achieves higher accuracy and precision, with minimal misclassifications across all obesity levels. In detail, it correctly identifies 88 instances of obesity level 0 with only two misclassifications, at obesity level 1, 86 instances correctly predicted with only seven misclassificatiuons, and similar results for other obesity levels, as shown in Figure 5f, showcasing its exceptional predictive capabilities. In contrast, models like CNN, RNN, LSTM, Bi-LSTM, and TabNet exhibit higher misclassification rates, particularly in distinguishing between different obesity levels. The Bi-LSTM model, for instance, misclassifies 18 instances of obesity level 1 as other obesity levels, as shown in Figure 5d, indicating a higher confusion between these classes. The TbaNet model outperforms other models, although its performance is slightly below the proposed ABi-LSTM model.

Moreover, the ABi-LASTM model demonstrates remarkable consistency in its predictions, with minimal variations in misclassifications. This consistency is vital for applications requiring reliable and uniform predictions. Furthermore, the ABi-LASTM model demonstrates a higher level of resilience, leading to a reduced occurrence of misclassifications across different classes when compared to alternative models. To recap, the analysis of the confusion matrix strongly affirms that the ABi-LASTM model outperforms traditional models such as CNN, RNN, LSTM, Bi-LSTM, and TabNet in predicting obesity levels. Its superior accuracy, stability, and robustness establish it as the preferred option for this particular task, bearing noteworthy implications for healthcare applications and predictive modeling in related fields.

#### 4.4.2. Assessing Model Effectiveness: Accuracy, Precision, Recall, and F1 Score

In our quest for precise obesity level prediction, we conducted a thorough assessment of a range of deep learning models, encompassing CNN, RNN, LSTM, Bi-LSTM, and TabNet. In this section, we present the performance metrics, which include Accuracy, Precision, Recall, and F1 Score, to evaluate the effectiveness of these models. In Table 4, we provide a detailed comparative analysis using different performance metrics for each of the models:

The results presented in Table 4 show the outstanding performance of our proposed ABi-LSTM model in predicting obesity levels. With an impressive Accuracy of 96.5%, our model showcases its ability to make accurate predictions.

Moreover, the ABi-LSTM model demonstrates a remarkable Precision score of 96.2%, signifying its precision in correctly classifying obesity levels. The Recall score of 95.9% emphasizes the model’s capacity to effectively identify true positive cases. The F1 Score, a harmonic mean of Precision and Recall, attains an exceptional 96.1%, reflecting the model’s overall balance in classification performance.

In comparison to the other state-of-the-art DL models tested, the ABi-LSTM model clearly outperforms them across all evaluated metrics. Notably, it surpasses the closest competitor, TabNet, by a clear margin, as shown in Figure 6.

The ABi-LSTM model’s Accuracy surpasses TabNet by 0.5%, Bi-LSTM by 3.3%, indicating its superior overall prediction accuracy. In terms of Precision, our model excels by 0.4% and 2.7% compared to TabNet and Bi-LSTM, respectively, underlining its precision in classifying obesity levels. The Recall rate of ABi-LSTM exceeds that of Bi-LSTM by 2.8%, indicating its ability to capture more true positive instances. Lastly, the F1 Score of ABi-LSTM outperforms TabNet and Bi-LSTM by 0.3% and 4.1%, respectively, showcasing its exceptional balance between Precision and Recall.

These results affirm the substantial performance advantages of the proposed ABi-LSTM model over existing DL approaches, making it a highly promising solution for accurate obesity level prediction.

## 5. Discussion

In the presented Table 5, we conduct a comprehensive evaluation of various machine learning and deep learning models used in the domain of obesity prediction, each offering a unique approach to this critical health issue. The models analyzed encompass diverse techniques, including Classification and Regression Trees (CART), Support Vector Machines (SVM), deep neural networks (DNNs), and Random Forest, reflecting the versatility of methods applied in addressing the problem of obesity prediction. Among the models evaluated, the work by Thamrin et al. [65] stands out as one of the pioneering studies. Their research explores the use of machine learning techniques, such as CART, Naïve-Bayes, and Logistic Regression, to classify individuals into obese and non-obese categories using the RISKESDAS 2018 dataset. The achieved accuracy of 79.8% suggests a reasonable level of predictive performance, although this study lacks certain advanced neural network architectures.

Furthermore, Montañez et al. [66] proposed an ML approach for obesity prediction based on publicly available genetic profiles. Leveraging SVM, they achieved an impressive accuracy of 90.5%. However, the precision and recall values are not reported, leaving room for a more comprehensive assessment of the model’s predictive power. Similarly, Kim et al. [67] tackle the challenge of predicting obesity risk from nutritional intake using the 4–7th Korea National Health and Nutrition Examination Survey (KNHANES). Their use of deep neural networks (DNNs), Logistic Regression, and Decision Tree models in a multi-class classification setting results in a moderate accuracy of 70.3%. Unfortunately, the reported precision, recall, and F1 Score values are not mentioned in the paper, making it challenging to assess the model’s performance fully.

In [68], Dugan et al. focus on early prediction of childhood obesity after age two using the CHICA dataset. They employ a Decision Tree model (ID3) and attain an accuracy of 85%. The model also exhibits competitive precision, recall, and F1 Score values of 84%, 89%, and 88%, respectively, suggesting a balanced performance. These notable studies in the obesity prediction domain set the stage for comprehensively evaluating our proposed ABi-LSTM model. The results are striking, with the ABi-LSTM model achieving an exceptional Accuracy of 96.5%. This places it firmly at the forefront of predictive accuracy in the field, surpassing all the models examined. Furthermore, the precision score of 96.2% signifies the model’s precision in correctly classifying obesity levels, while the recall score of 95.9% emphasizes its ability to identify true positive cases effectively. This demonstrates the model’s remarkable accuracy and proficiency in producing precise and comprehensive predictions. The F1 Score, an amalgamation of precision and recall, attains an outstanding value of 96.1%, underlining the model’s overall balance in classification performance. The proposed ABI-LSTM model’s performance is exceptional and cements its position as a top-tier solution for accurate obesity level prediction.

**Table 5 bioengineering-11-00533-t005:** Machine learning models for obesity prediction.

Ref	Research Goals	Data Source	Models Used	Classification Type	Accuracy (%)	Precision (%)	Recall (%)	F1 Score (%)
[65]	Predicting Obesity in Adults Using Machine Learning Techniques	RISKESDAS 2018	CART, Naïve-Bayes, Logistic Regression	Binary	79.8	69.56	—	71.49
[66]	Machine Learning Approaches for the Prediction of Obesity	DTCGT from PGP (NHGRI)	SVM	Binary	90.5	—	64.7	—
[67]	Classification and Prediction on the Effects of Nutritional Intake	KNHANES	DNN, Logistic Regression, Decision Tree	Multi-class	70.3	—	—	—
[69]	Machine Learning Approach for the Early Prediction of Obesity	UK’s Millennium Cohort Study (MCS)	Multilayer Perceptron	Binary	96	96	92	93.96
[70]	Obesity Prediction Using Ensemble Machine Learning Approaches	—	Ensemble ML Model	Binary	89.68	—	—	—
[68]	Machine Learning Techniques for Prediction of Early Childhood Obesity	CHICA	Decision Tree (ID3)	Binary	85	84	89	88
[71]	Using Machine Learning to Predict Obesity in High School Students	Biennial YRBSS	k-NN	Binary	88.82	—	—	—
[72]	A Hybrid Approach Based on Machine Learning to Identify the Causes of Obesity	ASFHC in Turkey	Hybrid of LR and LDT	Binary	91.4	94.9	90.4	90.4
[73]	Machine Learning Techniques to Predict Overweight or Obesity	Collected through a survey	Random Forest	Binary	78	79	78	78
Proposed Abi-LSTM	Obesity level prediction using advanced Bi-LSTM incorporating with Attention mechanism.	Obesity Levels & Life Style	ABi-LSTM	Multi-class	96.5	96.2	95.9	96.1

In a direct comparison with state-of-the-art deep learning models such as CNN, RNN, LSTM, Bi-LSTM, and TabNet, the ABi-LSTM model outperforms them across all evaluated metrics. The advantages are significant, with the ABi-LSTM model surpassing the closest competitor, Bi-LSTM, by substantial margins in terms of Accuracy, Precision, Recall, and F1 Score. In summary, the proposed ABI-LSTM model not only showcases exceptional accuracy but also excels in precision, recall, and the overall balance between these key performance measures. Its superior performance substantiates its role as a pioneering solution in the domain of obesity prediction, offering robust and precise predictive capabilities that surpass existing models in the field.

These compelling results underscore the substantial advancements of the proposed work that contribute to the field of obesity prediction, with far-reaching implications for healthcare and related research.

## 6. Conclusions

In this study, we have conducted an extensive analysis of various ML and DL models employed in obesity prediction, incorporating a wide array of methodologies, from tree-based models and support vector machines to deep neural networks and random forests. These multifarious approaches have made vital contributions to our understanding of the factors that are highly impacting obesity and have paved the way for the introduction of our novel ABi-LSTM model. The proposed ABi-LSTM model marks a significant advancement in the realm of obesity level prediction in multi-label classification problems. Additionally, as the global trend toward smart city initiatives gains momentum, our research plays a vital role in connecting healthcare with urban development. It illuminates how data-driven approaches can be harnessed to encourage healthier lifestyles within urban environments.

Achieving an impressive accuracy rate of 96.5%, ABi-LSTM outperforms all the existing frameworks we evaluated in this paper, showcasing an exceptional level of predictive precision. Furthermore, its Precision score of 96.2% highlights its capacity for making highly precise classifications, while the Recall score of 95.9% underscores its effectiveness in identifying true positive cases. The remarkable F1 Score of 96.1% further attests to the model’s overall balance in classification performance.

Comparing our ABi-LSTM model with state-of-the-art deep learning models such as CNN, RNN, LSTM, Bi-LSTM, and TabNet, it surpasses them across all key metrics, marking a significant breakthrough in the field of obesity prediction. The proposed model excels not only in predictive accuracy but also in precision and recall, reinforcing its position as an innovative solution.

In conclusion, our study underscores the substantial performance advantages of the ABi-LSTM model over existing deep learning approaches. Its exceptional precision, recall, and overall balance in these vital performance measures signify its robustness and effectiveness in predicting obesity levels. We firmly believe that this model holds profound implications for healthcare and related research, offering an exceptionally accurate tool for obesity level prediction.

## 7. Future Suggestions

As we look to the horizon, several exciting avenues for research beckon. Expanding the breadth of our model to incorporate a wider range of health-related data sources, including the integration of socio-economic factors and dietary habits, has the potential to significantly enhance predictive accuracy. Furthermore, with the rise of smart cities and the ever-increasing volume of data they generate, exploring the synergy between our model and the data streams from urban environments could be transformative. The inclusion of real-time data from smart city infrastructure offers the opportunity for continuous monitoring, enabling a dynamic approach to obesity prediction and prevention. In addition, applying our model to different demographic populations and diverse healthcare settings could provide invaluable insights into tailoring interventions and strategies. Future research might also address the intricate ethical and privacy considerations associated with using health data within the context of smart cities. These collective efforts will continue to advance our understanding of obesity prediction and its potential to enhance public health and patient care within the evolving landscape of smart cities.

## Figures and Tables

**Figure 1 bioengineering-11-00533-f001:**
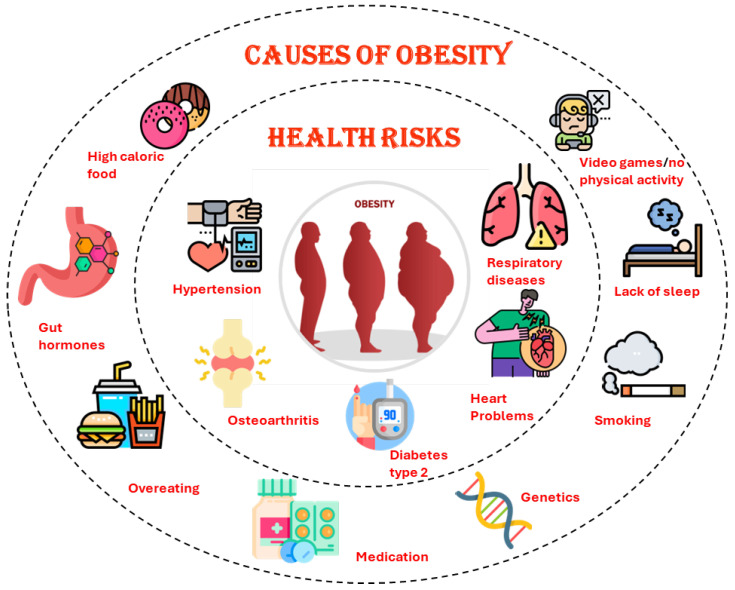
Causes of obesity in human body because of daily life routine.

**Figure 2 bioengineering-11-00533-f002:**
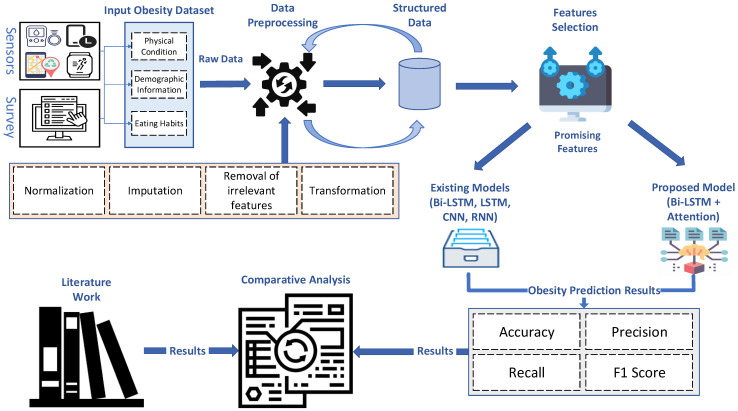
A detailed framework of the proposed ABi-LSTM.

**Figure 3 bioengineering-11-00533-f003:**
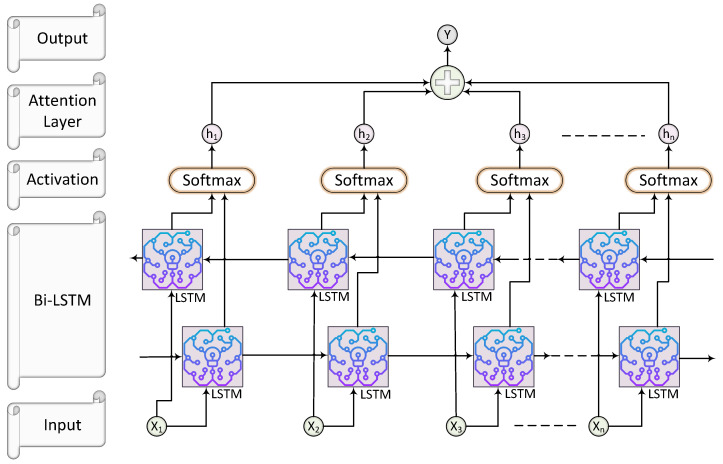
Proposed attention-based Bi-LSTM for obesity prediction.

**Figure 4 bioengineering-11-00533-f004:**
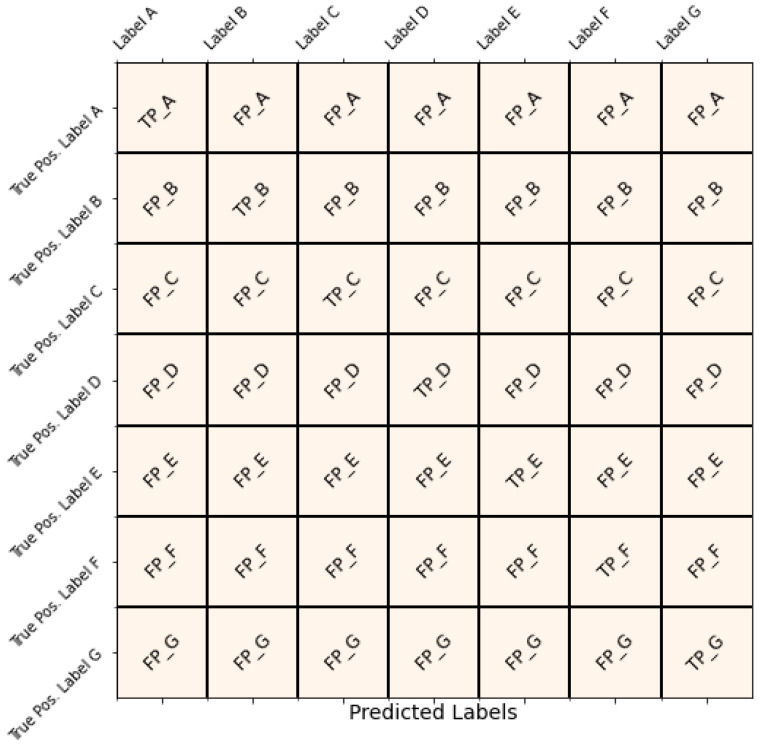
Evaluation of the obesity prediction experiments.

**Figure 5 bioengineering-11-00533-f005:**
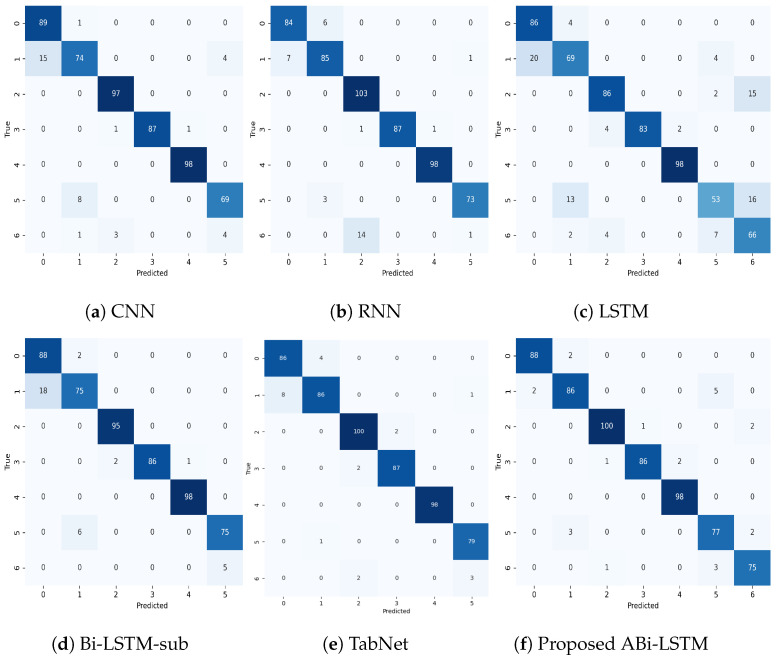
An in-depth analysis of the proposed ABi-LSTM with conventional DL models using a confusion matrix.

**Figure 6 bioengineering-11-00533-f006:**
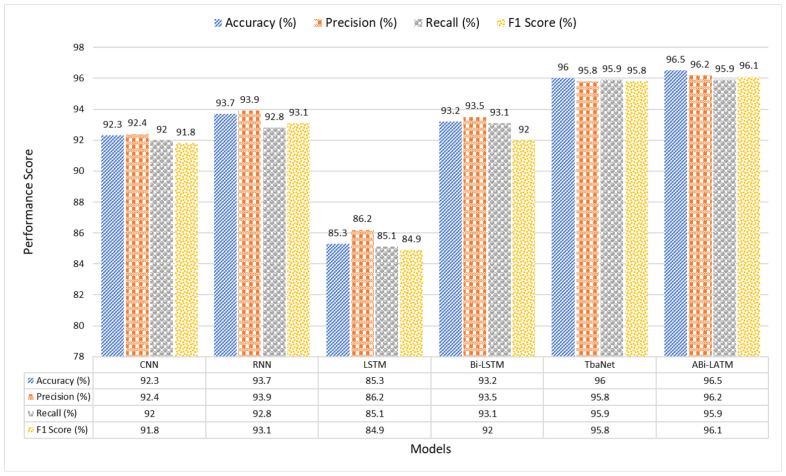
A visual illustration of the model results.

**Table 1 bioengineering-11-00533-t001:** A summary of the obesity data used in experiments.

Category	Feature	Description	Meaning
Eating Habits	FAVC	Frequent consumption of high-calorie food	Frequent consumption of high-calorie foods can lead to weight gain and obesity-related health issues, emphasizing the importance of moderating such intake for better health.
	FCVC	Frequency of consumption of vegetables	The frequency of consumption of vegetables is a crucial dietary aspect linked to overall health. Regularly consuming vegetables has numerous health benefits, including improved digestion, lower risk of chronic diseases, and weight management. It underscores the significance of incorporating a variety of vegetables into one’s diet to maintain a balanced and healthy lifestyle.
	NCP	Number of main meals	The number of main meals is pivotal in obesity. Irregular eating disrupts metabolism, affecting weight. Consistency in meals aids in weight control.
	CAEC	Consumption of food between meals	Consumption of food between meals influences obesity risk. Excessive snacking may lead to overconsumption, contributing to weight gain.
	CH2O	Consumption of water daily	Consumption of water daily plays a crucial role in managing obesity. Proper hydration can aid metabolism and control appetite, helping in weight management.
	CALC	Consumption of alcohol	Consumption of alcohol pertains to the amount and frequency of alcohol intake. Excessive alcohol consumption is linked to weight gain and can contribute to obesity, making it crucial to monitor and moderate alcohol consumption for a healthier lifestyle.
Physical Condition	SCC	Calories consumption monitoring	Calorie consumption monitoring involves keeping track of calorie intake. This awareness can be instrumental in managing weight and preventing obesity by ensuring a balanced diet.
	FAF	Physical activity frequency	Physical activity frequency refers to how often an individual engages in physical activities. Regular physical activity is essential for maintaining a healthy weight and preventing obesity, underscoring the importance of a consistent exercise routine in one’s lifestyle.
	TUE	Time using technology devices	Time using technology devices highlights how much time individuals spend using various gadgets such as smartphones, computers, and tablets. Excessive screen time can contribute to a sedentary lifestyle, which is associated with a higher risk of obesity. Therefore, monitoring and managing technology usage are essential aspects of a healthy lifestyle.
	MTRANS	Transportation used	Transportation choice, indicated by MTRANS, significantly impacts obesity rates. Reliance on sedentary modes like automobiles or public transportation often correlates with a higher risk of obesity due to reduced physical activity. Encouraging more active transportation methods can be a crucial strategy in obesity prevention.
Other Variables	Gender, Age, Height, Weight	—	Gender, age, height, and weight are fundamental variables in assessing and understanding obesity. These demographic and physiological factors play pivotal roles in determining an individual’s risk of obesity and contribute to the complexity of obesity-related research and interventions.

**Table 2 bioengineering-11-00533-t002:** System configuration and description.

System Components	Description
Operating System	Windows 10 for PC Server
Main Memory	64 GB RAM
Processor	12th Gen Intel(R) Core(TM) i9-12900K 3.20 GHz
Programming Language	Python 3
IDE	PyCharm Professional
Storage	MS Excel, MySQL
Core Libraries	Pandas, Scikit-Learn, Keras, TensorFlow, Seaborn, Matplotlib, etc.

**Table 3 bioengineering-11-00533-t003:** Configuration of attention-based Bi-LSTM model.

Parameter	Value
Input Dimension	16
Hidden Dimension	64
Number of Layers	3
Dropout Rate	0.4
Bidirectional	Yes
Attention Mechanism	Soft Attention
Attention Dimension	64
Activation Functions	Tanh
Learning Rate	0.001

**Table 4 bioengineering-11-00533-t004:** Model effectiveness: Accuracy, Precision, Recall, and F1 Score analysis.

Models	Accuracy (%)	Precision (%)	Recall (%)	F1 Score (%)
CNN	92.3	92.4	92.0	91.8
RNN	93.7	93.9	92.8	93.1
LSTM	85.3	86.2	85.1	84.9
Bi-LSTM	93.2	93.5	93.1	92.0
TabNet	96.0	95.8	95.9	95.8
ABi-LSTM	96.5	96.2	95.9	96.1

## Data Availability

The dataset for estimation of obesity levels based on eating habits and physical condition in individuals from Colombia, Peru and Mexico can be found at https://doi.org/10.1016/j.dib.2019.104344 (accessed on 1 April 2024).

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
