# Peer review of "Unraveling the Potential of Attentive Bi-LSTM for Accurate Obesity Prognosis: Advancing Public Health towards Sustainable Cities"

_bioengineering, 2024, doi:10.3390/bioengineering11060533_

Round 1

Reviewer 1 Report

Comments and Suggestions for Authors

The authors proposed an attention-based Bi-LSTM for accurate obesity prognosis. It is an interesting research topic. However, some things could be improved, and these are listed below.

1. In the abstract section, I suggest the author detail the proposed approach for the attention-based Bi-LSTM. The motivation and importance of this study should be improved.

2. In the method section, the authors only detail the process of a research. However, the readers would like to know the process of your approaches. What is the data format? What are the data types? How to process it? What is the structure of the proposed Attention-based Bi-LSTM? The attention-based Bi-LSTM is very important and should be detailed in the structure.

3. How the attention layer works? What type of attention layer?

4. The detailed parameters of the attention-based Bi-LSTM should be described.

5. The detailed information on the testing dataset should be described. For example: how many subjects? how many instances? The evaluation of the proposed approach should be detailed. For example: What are the training data and testing data?

6. Generally, the LSTM and RNN are suitable for sequence data. How to compare with the CNN and CART…? The experimental setup should be detailed.

7. What is the SOTA baseline system? The authors need to compare the SOTA systems under the same (almost) experimental conditions.

8. The authors need to reorganize this manuscript. It is very difficult to follow for readers.

Comments on the Quality of English Language

Moderate editing of English language required

Author Response

Thank you for your valuable feedback to our manuscript. We have revised the manuscript to address your concerns. We hope that the revised version of the manuscript is in much better shape now.

The authors proposed an attention-based Bi-LSTM for accurate obesity prognosis. It is an interesting research topic.

However, some things could be improved, and these are listed below.

  1. In the abstract section, I suggest the author detail the proposed approach for the attention-based Bi-LSTM. The motivation and importance of this study should be improved.

Thank you for your appreciation and the time you invested in our work and help us raise its quality. We sincerely appreciate your valuable feedback. In response to your suggestion, we have revisited the abstract section to provide a more detailed description of our proposed approach for the attention-based Bi-LSTM model.

We have enhanced the abstract to highlight the novel aspects of our approach, specifically integrating attention mechanisms with bidirectional Long Short-Term Memory (Bi-LSTM) networks. This integration allows our model to dynamically focus on relevant input features, capturing intricate dependencies and temporal patterns in longitudinal data, thereby enhancing both interpretability and predictive accuracy.

  1. In the method section, the authors only detail the process of a research. However, the readers would like to know the process of your approaches. What is the data format? What are the data types? How to process it? What is the structure of the proposed Attention-based Bi-LSTM? The attention-based Bi-LSTM is very important and should be detailed in the structure.

Thank you for your constructive feedback. We have reorganized and enriched the methodology section to ensure clarity and accessibility for readers. By incorporating the suggested details, such as the data format, types, and processing methods (section 3.1), causes and effects of obesity (section 3.2), and detailed description of proposed model (section 3.3), we aim to provide readers with a comprehensive understanding of our approach, facilitating easier comprehension of the core ideas and important aspects of our research.

  1. How the attention layer works? What type of attention layer?

Thank you for your constructive feedback, which enhances the approach discussed. The attention layer in our proposed model dynamically assigns weights to input features based on their relevance to predicting obesity, focusing on key features such as height, weight, and physical activity levels. We utilize a soft attention mechanism, which calculates attention weights using a learned function based on input-context vector similarity. This mechanism enhances our model's ability to capture complex relationships within the data. We incorporate the comprehensive mechanism in the proposed framework.   

  1. The detailed parameters of the attention-based Bi-LSTM should be described.

We appreciate the opportunity to elaborate on the parameters of our attention-based Bi-LSTM model. In our paper, we added a comprehensive description of the key parameters that define the architecture and configuration of our model. These parameters, including input dimension, hidden dimension, number of layers, dropout rate, bidirectional nature, attention mechanism type, attention dimension, activation functions, and learning rate, are meticulously chosen to address the complexities of our multilabel classification task, specifically in predicting obesity. We believe that detailing these parameters is essential for understanding how our model is optimized to process the features relevant to obesity prediction effectively and to capture intricate patterns and relationships within the data.

  1. The detailed information on the testing dataset should be described. For example: how many subjects? how many instances? The evaluation of the proposed approach should be detailed. For example: What are the training data and testing data?

We acknowledge your attention to detail. The manuscript has been updated to include the data split details and other relevant information pertaining to the dataset, particularly in section 4.4. We trust that this addresses your concern, and we sincerely appreciate your contribution to enhancing the clarity of our work.

  1. Generally, the LSTM and RNN are suitable for sequence data. How to compare with the CNN and CART…? The experimental setup should be detailed.

Thank you for your feedback on an important aspect of our implementation. We utilized LSTM and RNN architectures, which excel in processing data sequences and are suitable for tasks involving time series or sequential data, such as our obesity level prediction problem. CNN model cannot capture sequential information, so CNN didn't perform well. CART models, known for their simplicity and interpretability, were included in our comparison to provide a baseline for assessing the performance of more complex deep learning architectures. While CART models cannot capture temporal dependencies like LSTM and RNN, they offer transparency and ease of interpretation, making them valuable for understanding the underlying decision-making process. The experimental setup is described in Section 4.4, and System configuration is defined in Section 4.1.

  1. What is the SOTA baseline system? The authors need to compare the SOTA systems under the same (almost) experimental conditions.

Thank you for your kind feedback. Our study established a comprehensive baseline by comparing the proposed ABi-LSTM model with existing state-of-the-art (SOTA) systems commonly used for obesity level prediction. These SOTA systems encompass a variety of deep learning and traditional machine learning models, including Convolutional Neural Networks (CNNs), Recurrent Neural Networks (RNNs), Long Short-Term Memory (LSTM) networks, and traditional machine learning algorithms such as Classification and Regression Trees (CART) and Support Vector Machines (SVMs). We meticulously structured our setup according to Unified Dataset, Adhering to Standardized Experimental Protocol with Consistent Evaluation Metrics to ensure a fair comparison under almost identical experimental conditions.

  1. The authors need to reorganize this manuscript. It is very difficult to follow for readers.

Thank you for your feedback. We have taken your suggestion seriously and reorganized the manuscript to enhance readability and comprehension for readers. We have clarified the research problem and provided a comprehensive overview of the obesity epidemic, including previous prediction techniques and recent advancements in deep learning models for medical image analysis in the introduction section.

We have organized the literature review section to provide a systematic analysis of various approaches to obesity prediction, highlighting insights and limitations of existing models. This section now includes discussions on recent advancements in transformer-based architectures and novel attention mechanisms.

The methodology section has been revised to provide a clear and detailed description of our proposed approach, the Attention-based Bi-LSTM (ABi-LSTM) model, including data preprocessing steps, model architecture, and training procedures.

We have presented the results of our experiments in a structured and easy-to-follow manner, including performance metrics and comparisons with existing models.

The discussion section now offers a comprehensive analysis of the results, discussing the implications of our findings and addressing the limitations of our approach.

We have thoroughly revised all the manuscript sections based on your and other reviewers' valuable comments. We would like to thank you for investing your time and efforts in reading the manuscript and providing feedback. We hope that the revised manuscript is in much better shape now!

Reviewer 2 Report

Comments and Suggestions for Authors

The authors present a comprehensive analysis of existing ML and DL models in the obesity prediction domain and introduce a hybrid model for obesity level prediction. It is interesting. However, there some major concerns must be addressed.

1.      The prediction accuracy is commonly used in classification. Yet, the MSE is always used in prediction tasks. Should clarify it.

2.      What is the motivation for proposing the ABi-LSTM? What the problem you want to address in the application or technology? I do not find it anywhere.

3.      The authors should state the research problem in the abstract. Next, should introduce it in the introduction section. Finally, should analyze it in the discussion section. In addition, do not state the research problem at a fuzzy level, yet at a small direction.

4.      Why introduce the attention to Bi-LSTM? Should clarify it.

5.      There are some Transformer and novel Attention-based CNN commonly used in medical and health information processing. The authors should analyze those in the introduction or discussion, for example, The Swin-Transformer network based on focal loss is used to identify images of pathological subtypes of lung adenocarcinoma with high similarity and class imbalance, LA-ViT: A Network with Transformers Constrained by Learned-Parameter-Free Attention for Interpretable Grading in a New Laryngeal Histopathology Image Dataset.

6.      The end-to-end optimizing network already become mainstream methods. Why use an extracting feature, i.e., a two-stage method? What is the advantage?

7.      The authors do not present how to split the dataset for training, testing, and validation. Should present it.

8.      The author should compare some new Transformer and CNN model in the experiment.

Comments on the Quality of English Language

None

Author Response

Thank you for your valuable feedback to our manuscript. We have revised the manuscript to address your concerns. We hope that the revised version of the manuscript is in much better shape now.

The authors present a comprehensive analysis of existing ML and DL models in the obesity prediction domain and introduce a hybrid model for obesity level prediction. It is interesting.

However, there some major concerns must be addressed.

  1. The prediction accuracy is commonly used in classification. Yet, the MSE is always used in prediction tasks. Should clarify it.

We primarily focus on a classification task rather than a prediction task. We aim to classify individuals into different obesity levels based on various features. Therefore, the metric of choice for evaluating the performance of our models is classification accuracy rather than mean squared error (MSE), which is commonly used in regression or prediction tasks. The loss mentioned was used during the training and only for assessing performance. We updated our paper according to the standard classification task for more enhancement.

  1. What is the motivation for proposing the ABi-LSTM? What the problem you want to address in the application or technology? I do not find it anywhere. (Reviewer 1 but Tech Answer)

Thank you for your feedback on an important aspect. As mentioned in the Proposed framework description, we utilized ABi-LSTM, which excels in processing data sequences and is suitable for tasks involving sequential data, such as our obesity level prediction problem. As a variant of a recurrent neural network (RNN), the ABi-LSTM model is well-suited to handle sequential data, like the obesity dataset we used. The ability to proficiently capture temporal dependencies and patterns within the data is crucial for comprehending the intricate linkages that exist between obesity levels and lifestyle factors throughout time.

As discussed in the Introduction section, Our research aims to address the critical issue of accurately predicting obesity levels in individuals. Obesity is a multifaceted health problem with significant implications for public health and individual well-being. Accurate prediction of obesity levels can facilitate early intervention and personalized healthcare strategies, ultimately leading to better health outcomes.

  1. The authors should state the research problem in the abstract. Next, we should introduce it in the introduction section. Finally, should analyze it in the discussion section. In addition, do not state the research problem at a fuzzy level, yet at a small direction.

   Thank you for your valuable feedback. Our introduction section aims to provide a thorough background on the obesity epidemic and the various prediction techniques employed in previous research. We introduce our proposed solution, the Attention-based Bi-LSTM (ABi-LSTM) model, as a novel approach to addressing the limitations of existing methods.

    In response to the reviewer's suggestion, we have structured our paper to address the research problem at a specific level by outlining the challenges associated with obesity prediction and the need for improved models. We introduce the ABi-LSTM model as our contribution to this field, highlighting its potential to significantly enhance predictive performance.

Furthermore, our discussion section analyzes the results obtained from applying the ABi-LSTM model, evaluating its accuracy, precision, recall, and F1 score compared to existing state-of-the-art models. We also discuss the implications of our findings for public health and related research areas, emphasizing the importance of comprehensive data collection and the potential of data-driven techniques to promote healthy living in urban environments.

  1. Why introduce the attention to Bi-LSTM? Should clarify it.

Thank you for asking about this. Introducing attention to Bi-LSTM enhances its performance by allowing the model to focus on the most relevant parts of the input sequence. This mechanism helps capture important features and patterns, particularly in tasks with varying significance levels across the input data. By dynamically allocating attention, the model can better understand the temporal relationships and dependencies within the sequence, leading to more accurate predictions. Additionally, attention mitigates the vanishing gradient problem and improves the model's ability to learn long-range dependencies, making it a powerful tool for sequence modeling tasks like obesity level prediction.

  1. There are some Transformer and novel Attention-based CNN commonly used in medical and health information processing. The authors should analyze those in the introduction or discussion, for example, The Swin-Transformer network based on focal loss is used to identify images of pathological subtypes of lung adenocarcinoma with high similarity and class imbalance, LA-ViT: A Network with Transformers Constrained by Learned-Parameter-Free Attention for Interpretable Grading in a New Laryngeal Histopathology Image Dataset.

Thank you for your suggestion. We acknowledge the significance of recent advancements in deep learning models, particularly transformer-based architectures and novel attention mechanisms, in medical image analysis. In the related work section, we have now included analysis and discussion on prominent models such as the Swim-Transformer network and LA-ViT. We believe that by incorporating insights from these models, our study aims to contribute to the ongoing advancements in medical and health information processing. 

  1. The end-to-end optimizing network already become mainstream methods. Why use an extracting feature, i.e., a two-stage method? What is the advantage?

The feature extraction process isolates pertinent features related to obesity within datasets cluttered with diverse information. It ensures that only relevant attributes are utilized for effective analysis and model training.

  1. The authors do not present how to split the dataset for training, testing, and validation. Should present it.

We appreciate your attention to detail. The data split details and other information related to data have been incorporated into the manuscript, specifically in section 4.4. We hope this addresses your concern, and we thank you for helping us improve the clarity of our work.

  1. The author should compare some new Transformer and CNN model in the experiment.

Thank you for your valuable feedback. In response to your suggestion, we have included some new models in our experiment and updated the results accordingly. Additionally, we have provided a comparative analysis that incorporates these new models, offering a more comprehensive understanding of their performance compared to our proposed model.

We have thoroughly revised all the manuscript sections based on your and other reviewer valuable comments. In the end, we would like to thank you for investing your time and efforts in reading the manuscript and your feedback comments. We hope that the revised manuscript is in much better shape now!

Round 2

Reviewer 1 Report

Comments and Suggestions for Authors

All issues that I am concerned about have been carefully considered. I suggest that it can be accepted.

Comments on the Quality of English Language

All issues that I am concerned about have been carefully considered. I suggest that it can be accepted.

Reviewer 2 Report

Comments and Suggestions for Authors

The authors have addressed my concerns well.

Comments on the Quality of English Language

None